Lighten up the dark: metazoan parasites as indicators for the ecology of Antarctic crocodile icefish (Channichthyidae) from the north-west Antarctic Peninsula

Kuhn Thomas 1
Zizka Vera M.A. 2
Münster Julian 1 muenster@em.uni-frankfurt.de
Klapper Regina 1
Mattiucci Simonetta 3
Kochmann Judith 1
Klimpel Sven 1
1 Institute for Ecology, Evolution and Diversity, Johann Wolfgang Goethe University, Senckenberg Biodiversity and Climate Research Centre, Senckenberg Gesellschaft für Naturforschung , Frankfurt am Main , Germany
2 Aquatic Ecosystem Research, Faculty of Biology, University of Duisburg-Essen , Essen , Germany
3 Department of Public Health and Infectious Diseases, Section of Parasitology, University of Roma “La Sapienza” , Rome , Italy
Baird Donald
Electronic publication date: 2018 May 11
Publication date: 2018
Volume: 6
Electronic Location ID: e4638
Received 2018 Jan 23; Accepted 2018 Mar 28
Copyright: © 2018 Kuhn et al.
Copyright year: 2018
Copyright holder: Kuhn et al.
License: This is an open access article distributed under the terms of the Creative Commons Attribution License, which permits unrestricted use, distribution, reproduction and adaptation in any medium and for any purpose provided that it is properly attributed. For attribution, the original author(s), title, publication source (PeerJ) and either DOI or URL of the article must be cited.
License URL: https://creativecommons.org/licenses/by/4.0/

Keywords: Channichthyidae, Anisakid nematodes, Contracaecum, Pseudoterranova, Champsocephalus gunnari, Chaenodraco wilsoni, Neopagetopsis ionah, Pagetopsis macropterus, Pseudochaenichthys georgianus, Antarctica

Funding: The authors received no funding for this work.

==============================
Due to its remote and isolated location, Antarctica is home to a unique diversity of species. The harsh conditions have shaped a primarily highly adapted endemic fauna. This includes the notothenioid family Channichthyidae. Their exceptional physiological adaptations have made this family of icefish the focus of many studies. However, studies on their ecology, especially on their parasite fauna, are comparatively rare. Parasites, directly linked to the food chain, can function as biological indicators and provide valuable information on host ecology (e.g., trophic interactions) even in remote habitats with limited accessibility, such as the Southern Ocean. In the present study, channichthyid fish (Champsocephalus gunnari: n = 25, Chaenodraco wilsoni: n = 33, Neopagetopsis ionah: n = 3, Pagetopsis macropterus: n = 4, Pseudochaenichthys georgianus: n = 15) were collected off South Shetland Island, Elephant Island, and the tip of the Antarctic Peninsula (CCAML statistical subarea 48.1). The parasite fauna consisted of 14 genera and 15 species, belonging to the six taxonomic groups including Digenea (four species), Nematoda (four), Cestoda (two), Acanthocephala (one), Hirudinea (three), and Copepoda (one). The stomach contents were less diverse with only Crustacea (Euphausiacea, Amphipoda) recovered from all examined fishes. Overall, 15 new parasite-host records could be established, and possibly a undescribed genotype or even species might exist among the nematodes.

Introduction

The exploration of remote habitats such as those of Antarctica has significantly influenced the understanding of the molecular, physiological, and behavioral mechanisms by which organisms adapt in order to survive, grow, and reproduce under extreme conditions. The evolutionary origin of Antarctic crocodile icefish (Notothenioidei) can be dated back to the isolation of the Antarctic continent and the surrounding Southern Ocean (Oligocene/Miocene boundary, approx. 25 mya) by the opening of the Drake Passage and the formation of the Antarctic circumpolar current (Eastman, 1993; Near, Pesavento & Cheng, 2003; Kock, 2005; Garofalo et al., 2009). This isolation gave rise to a roughly circular ocean front, the Antarctic Polar Front, which acts as a climatic and biologic barrier and decouples the Southern Ocean surrounding Antarctica from warm subtropical waters further north. The extensive temperature decrease coupled with ice sheet expansion, following the opening of the Drake Passage, led to a process of selection by exposure to freezing conditions in resident communities. The extinction of a large part of the endemic fish fauna due to freezing intolerance fueled the evolution of cold-adapted species (Bargelloni et al., 1994; Wöhrmann, 1997; Kock, 2005; Cheng, di Prisco & Verde, 2009; Near et al., 2012). This period of extreme cooling, ice sheet growth, and formation of sea ice along the Antarctic continental shelf, together with little competition between species, is thought to be the period of strong diversification and radiation within the Notothenioidei (approx. 15 mya) (Eastman, 1993). The suborder evolved from a benthic living ancestor endemic to Antarctica, which adapted to the cold environment by the expression of antifreeze glycoproteins, finally radiated, and successfully occupied many ecological niches (Bargelloni et al., 1994; Near, 2004; Kock, 2005). Today, the Notothenioidei strongly dominate the teleost fish diversity and abundance in the Southern Ocean and account for 90% of the fish biomass and for over 50% of all fish species in most regions of the Antarctic Sea (Wöhrmann, 1997; Flores et al., 2004; Kock, 2005; Reid et al., 2007; Near et al., 2012). The suborder contains 130 species in eight families (Artedidraconidae, Bovichtidae, Pseudaphritidae, Eleginopsidae, Nototheniidae, Harpagiferidae, Bathydraconidae, Channichthyidae) (Near, Pesavento & Cheng, 2003; Near et al., 2012).

Within the Notothenioidei, species of crocodile icefish (Channichthyidae) form a monophyletic group and are, with regard to morphological characters, one of the most derived clades of the Notothenioidae (Iwami, 1985; Near, Pesavento & Cheng, 2003; Kock, 2005). While Channichthyidae have been the focus of many studies dealing with their unique physiological adaptations (e.g., their complete lack of the oxygen-binding hemoglobin) (Sidell et al., 1997; Feller & Gerday, 1997; Garofalo et al., 2009), studies on their ecology are comparatively rare. Parasites can function as biological indicators as they are directly linked to the food chain and can provide valuable information on the ecology of the host. For instance, metazoan parasites, and especially helminths, have evolved complex life-cycles, including several hosts among different trophic levels, and are therefore deeply embedded within food webs. As a proxy for long-term trophic interactions, their analyses increase the knowledge on the ecology of the hosts and their linkage in the food web (Klimpel, Seehagen & Palm, 2003; Lafferty et al., 2008) without the need for great sampling efforts of the various trophic levels involved.

In the present study, we aimed to analyze the parasite fauna and stomach contents of Champsocephalus gunnari Lönnberg, 1905, Chaenodraco wilsoni Regan, 1914, Neopagetopsis ionah Nybelin, 1947, Pagetopsis macropterus Boulenger, 1907, and Pseudochaenichthys georgianus Norman, 1937, all members of the family Channichthyidae. The effort was made to increase our currently limited knowledge of their ecology and the host range of their associated parasite fauna.

Materials and Methods

A total of 80 channichthyid fish (C. gunnari: n = 25, C. wilsoni: n = 33, N. ionah: n = 3, P. macropterus: n = 4, P. georgianus: n = 15) were collected and sorted on board of the German RV Polarstern during the research cruise ANT-XXVIII/4 (2012) to the South Shetland Islands region, Elephant Island and the tip of the Antarctic Peninsula (CCAML statistical subarea 48.1) (see Table 1; Fig. 1). Fishes were caught with a bottom trawl (mouth opening: 2.5–3.2 × 16–18 m) with a towing time of 30 min. After each haul fish species were identified according to Gon & Heemstra (1990) and sorted by species. Individuals were packed in plastic bags, labelled and deep frozen at −40 °C for subsequent examination.

Table 1 Trawl and catch data.

Nr	Station	Date	Action	PosLat	PosLon	Depth (m)	C.w.	C.g.	N.i.	P.m.	P.g.	
1	PS79/204	March 20, 2012	Set	61°14.35′S	55°48.88′W	131.8	1				2	
Haul	61°16.06′S	55°47.31′W	146.9				
2	PS79/242	March 27, 2012	Set	61°35.90′S	57°17.00′W	430.0			2			
Haul	61°35.40′S	57°20.60′W	430.0	
3	PS79/247	March 28, 2012	Set	62°29.27′S	61°24.79′W	122.0	2					
Haul	62°28.18′S	61°22.00′W	122.0	
4	PS79/257	March 30, 2012	Set	62°01.50′S	59°36.20′W	176.0					2	
Haul	62°00.50′S	59°37.60′W	176.0	
5	PS79/259	March 30, 2012	Set	62°00.00′S	59°14.80′W	131.0					3	
Haul	62°00.00′S	59°10.70′W	131.0	
6	PS79/265	March 31, 2012	Set	61°49.30′S	58°34.70′W	193.0			1		1	
Haul	61°50.10′S	58°30.70′W	193.0	
7	PS79/268	April 1, 2012	Set	62°15.61′S	55°18.44′W	366.1	29					
Haul	61°13.77′S	55°17.68′W	351.9					
8	PS79/269	April 1, 2012	Set	62°21.40′S	55°18.90′W	314.0				4		
Haul	62°23.20′S	55°17.60′W	314.0	
	PS79/209	February 21, 2012	Set	n.d	n.d	n.d					3	
Haul	n.d	n.d	n.d	
	PS79/220	March 23, 2012	Set	n.d	n.d	425.0					1	
Haul	n.d	n.d	395.0	
	PS79/221	March 23, 2012	Set	n.d	n.d	n.d	1	25				
Haul	n.d	n.d	n.d				
	PS79/253	March 29, 2012		Course to PS79/269	300.0					2	
	PS79/266	March 30, 2012	Set	n.d	n.d	n.d					1	
Haul	n.d	n.d	n.d					
Notes:

Information on date, position (S = south, W = west) and depth (m) are given for each station. For some trawls data about the duration or position were missing (n.d). The number of individuals caught [n] is given for each species.

C.g., Champsocephalus gunnari; C.w., Chaenodraco wilsoni; N.i., Neopagetopsis ionah; P.m., Pagetopsis macropterus; P.g., Pseudochaenichthys georgianus.

Figure 1 Sampling locations off the Antarctic Peninsula.

Specific information on coordinates of the stations (1–8) are summarized in Table 1. Map data © Esri, DeLorme, GEBCO, NOAA, NGDC.

Parasitological examination and stomach content analysis

Total length and standard length were measured to the nearest millimeter and total weight was measured to the nearest 1.0 g. Gills, nostrils, skin, fins, eyes, and mouth cavity were examined for ectoparasites. Fish were then opened, the intestinal organs were removed, and transferred separately into petri dishes filled with 0.9% saline solution. Liver (LG), gonads (GO), full (SW), and empty stomachs (SWempty) were weighed to the nearest 0.1 g. The liver, gonads, stomach, gut, gall bladder, pyloric caeca were then dissected and microscopically examined for endoparasites with a magnification of 6.7–45× (SZ 61; Olympus, Hamburg, Germany). Host tissue was removed from isolated parasites. Digenea, Cestoda, Crustacea, and some nematodes were fixed in 4% borax buffered formalin and preserved in 70% ethanol with 4% glycerol for morphological identification. The remaining nematodes were preserved in EtOH (abs.) for subsequent molecular identification. Stomach contents were removed from each fish and isolated food items were separated and identified to the lowest possible taxon. Terminology for parasitological and ecological examination followed Bush et al. (1997). Parasite species were morphologically identified using original description as well as identification keys (Zdzitowiecki, 1991a; Klöser & Plötz, 1992; Zdzitowiecki, 1997; Rocka, 2004; Utevsky, 2005). In addition, randomly chosen subsamples from nematode specimens were identified using molecular methods.

Molecular nematode identification

Genomic DNA was isolated and purified from 117 parasite larvae for nematode species identification, using the peqGOLD Tissue DNA MicroSpin Kit (Peqlab Biotechnology GmbH, Erlangen, Germany) according to the instructions of the manufacturer. An approx. 639 bp long fragment of the mitochondrial cytochrome oxidase 2 (cox2) region was amplified using the primers 211-F (5′-TTT TCT AGT TAT ATA GAT TGR TTY AT-′3) and 210-R (5′-CAC CAA CTC TTA AAA TTA TC-3′) (Nadler & Hudspeth, 2000). The PCR-reaction (25 μl) included 12.5 μl Master-Mix (Peqlab Biotechnology GmbH, Erlangen, Germany) containing dNTP (0.4 mM), MgCl2 (4 mM), Buffer (40 mM Tris-HCL, 32 mM (NH4)2SO4, 0.02% Tween 20) HotStart Taq-Polymerase (0.625 u), 1.5 μl of each primer (10 pmol μl−1), 7 μl ddH2O, and 2.5 μl genomic DNA. The PCR temperature and cycling parameters were: initial denaturation at 94 °C for 180 s, 40 cycles of 94 °C for 30 s (denaturation), 46 °C for 60 s (annealing), 72 °C for 90 s (extension), followed by a final extension at 72 °C for 10 min. Samples without DNA were included as negative controls in each PCR run. PCR products were visualized on 1% agarose gels. To estimate the size of the PCR products, a 100 bp ladder marker (peqGOLD, Erlangen, Germany) was used. Afterwards, PCR products were purified with the peqGOLD Cycle-Pure Kit (Peqlab Biotechnology GmbH, Erlangen, Germany) following the instructions of the manufacturer. The purified products were sequenced by Seqlab (Goettingen GmbH, Germany) using primer 210-R.

Sequence analyses

Analyses were performed using Geneious v.8.1.7. Sequence data were compared with previously published Genbank data using a BLASTn search (Altschul et al., 1990). For phylogenetic reconstruction, a multiple sequence alignment including the 117 obtained sequences as well as 19 previously annotated reference sequences of phylogenetically closely related anisakid species was generated using MUSCLE as implemented in Geneious v.8.1.7 using default parameters (Alignment S3).

Bayesian inference analysis was performed on the 634 base pair MUSCLE alignment using MrBayes v3.1.2 (Huelsenbeck & Ronquist, 2001) as implemented in Geneious v.8.1.7. The GTR substitution model with gamma rate variation (five categories) was used. Default parameters were used for Markov Chain Monte Carlo parameters (chain length: 1,100,000; subsampling frequency: 200; heated chains 4; heated chain temp: 0.2 burn-in length: 10,000; random seed: 31,049). A consensus tree was constructed from the tree output files after discarding the first 10% (support threshold: 50%; burn-in: 10%) and visualised using TreeGraph 2 (Stöver & Müller, 2010). Anisakis simplex (s.s.) (Accession No: KT852484) as a common and closely related taxon was included as an outgroup.

Data analysis/statistics

Statistical analyses were conducted in R (version 3.0.2). Host biometric values, parasite abundances, and parasite intensities are given with standard deviation (±SD). Parasite fauna diversity was calculated using the Shannon–Wiener index (H′) and Pielous’s evenness (J′) according to Magurran (1988).

In order to compare the parasitological findings of the examined specimens with other members of the family Channichthyidae, data of metazoan parasites from the different species of this family were collected by a search on Google Scholar and Web of Knowledge. The names of the known channichthyid species, combined with the keywords “parasite” “Digenea,” “Monogenea,” “Cestoda,” “Nematoda,” “Acanthocephala,” and “Crustacea” were used. In addition to original publications, Klimpel et al. (2009) and Oguz et al. (2015) were taken into consideration. Validity of species names were checked by using The World Register of Marine Species (www.marinespecies.org). Only unambiguous records were included. The results are given in Table S1.

Results

Parasite infection data

A total of 2,563 individual parasites was isolated from the 80 samples of channichthyid fishes belonging to 14 different genera and 15 species, including 15 new host records (Tables 2 and 3; Table S2).

Table 2 Parasitological data from the examined channichthyid P. georgianus.

Species	Parasite	Stage	Location	n	P[%]	MI (I)	MA	H′	J′	
Pseudochaenichthys georgianus (n = 15)	Digenea		SP	7	20.0	2.3	0.5	1.5	0.6	
 Elytrophalloides oatesi	A	S	1	13.3	2.5 (1–4)	0.3	
 Gonocerca phycidis	A	S	5	6.7	1.0 (1)	0.1	
 Neolebouria antarctica	A	P	1	6.7	1.0 (1)	0.1	
Nematoda		BC/L/S/I/P/FT	460	93.3	32.8	30.7	
 Ascarophis nototheniae*	A	S	1	6.7	1.0 (1)	0.1	
 Contracaecum osculatum (s.l.)*	Lv	BC/L/S/I/P/FT	240	93.3	17.1 (2–54)	16.0	
 Contracaecum radiatum*	Lv	BC/L/S/I/P	171	86.7	13.2 (1–34)	11.4	
 Contracaecum sp.	Lv	BC/L/S/P	13	42.9	2.2 (1–4)	0.9	
 Nematoda indet	Lv	L/I	1	13.3	1.0 (1)	0.1	
 Pseudoterranova decipiens (s.l.)	Lv	BC/L/S	33	33.3	6.6 (2–16)	2.2	
Cestoda		BC/L/S/I/P/FT	1,774	100	118.3	118.3	
 Diphylloboththriidea indet	Lv	BC/L/S/I/P/FT	612	80.0	51.0 (1–170)	40.8	
 Tetraphyllidae indet	Lv	BC/L/S/I/P/FT	1,162	100	77.5 (4–238)	77.5	
Acanthocephala		BC/S/P	5	20.0	1.7	0.3	
 Corynosoma bullosum*	Lv	BC/S/P	5	20.0	1.7 (1–3)	0.3	
Hirudinae		SU/MC/G	8	33.3	1.6	0.5	
 Nototheniobdella sawyeri*	A	MC/G	6	26.7	1.5 (1–3)	0.4	
 Trulliobdella capitis	A	SU	2	13.3	1.0 (1)	0.1	
Copepoda		SU	1	6.7	1.0	0.0	
 Eubrachiella antarctica*	A	SU	1	6.7	1.0 (1)	0.1	
Notes:

A, adult; Lv, larval; BC, body cavity; L, liver; S, stomach; I, intestine; P, pylorus; FT, fat tissue; SU, surface; MC, mouth cavity; G, gills; n, total number of parasites; Location, location of the parasite inside the host; P[%], prevalence; MI, mean intensity; I, intensity (in parentheses); MA, mean abundance; H′, diversity index, J′, eveness.

Parasites isolated from stomach and intestine were located in the lumen of these organs. New host records are marked with an asterisk (*).

Table 3 Parasitological data from the examined channichthyids Chaenodraco wilsoni, Champsocephalus gunnari, Neopagetopsis ionah, and Pagetopsis macropterus.

Species	Parasite	Stage	Location	n	P[%]	MI (I)	MA	H′	J′	
Chaenodraco wilsoni (n = 33)	Digenea		S/I/P	3	6.1	1.5	0.1	1.9	0.8	
 Neolebouria antarctica	A	S/I/P	3	6.1	1.5 (1–2)	0.1	
Nematoda		L/S/I/P	58	48.5	3.9	1.9	
 Contracaecum osculatum (s.l.)*	Lv	L	35	39.4	2.7 (1–7)	1.1	
 Contracaecum radiatum*	Lv	L/S/I/P	17	33.3	1.5 (1–3)	0.5	
 Contracaecum sp.	Lv	L	6	3.0	6.0 (6)	0.2	
 Nematoda indet	Lv	L/P	5	12.1	1.3 (1–2)	0.2	
Cestoda		L/F/T	10	30.3	2.5	0.3	
 Diphyllobothriidea indet	Lv	L/FT	10	30.3	2.5 (1–4)	0.3	
Hirudinea		S/MC/G	43	30.3	1.4	0.4	
 Notobdella nototheniae*	A	S	32	15.2	6.4 (1–20)	1.0	
 Nototheniobdella sawyeri	A	MC/G	9	21.2	1.3	0.3	
 Trulliobdella capitis	A	S	2	6.1	1.0 (1)	0.1	
Champsocephalus gunnari (n = 25)	Digenea		P	2	8.0	1.0	0.1	1.6	0.9	
 Macvicaria georgiana*	A	P	2	8.0	1.0 (1)	0.1	
Nematoda		BC/L/S/I/P	51	72.0	2.8	2.0	
 Contracaecum osculatum (s.l.)*	Lv	L/P	14	40.0	1.4 (1–2)	0.6	
 Contracaecum radiatum*	Lv	L/I/P	10	28.0	1.4 (1–3)	0.4	
 Contracaecum sp.	Lv	P	5	16.0	1.2 (1–2)	0.2	
 Nematoda indet	Lv	BC/L/S/I/P	22	48.0	1.8 (1–6)	0.9	
Cestoda		L/S/P/FT	9	28.0	1.3	0.4	
 Diphyllobothriidea indet	Lv	L/S/P/FT	9	28.0	1.3 (1–2)	0.4	
Neopagetopsis ionah (n = 3)	Nematoda		L/S/P	9	100	3.0	3.0	1.2	0.5	
 Contracaecum osculatum (s.l.)*	Lv	L	6	100	2.0 (1–3)	2.0	
 Contracaecum radiatum*	Lv	P	2	66.7	1.0 (1)	0.7	
 Contracaecum sp.	Lv	S	1	33.3	1.0 (1)	0.3	
Cestoda		BC/L/S	5	66.7	2.5	1.7	
 Diphyllobothriidea indet	Lv	BC/L/S	5	66.7	2.5 (2–3)	1.7	
Pagetopsis macropterus (n = 4)	Nematoda		BC/L/S/I/FT	44	100	11.0	11.0	1.0	0.7	
 Contracaecum osculatum (s.l.)*	Lv	BC/L/S/FT	34	100	8.5 (3–19)	8.5	
 Contracaecum radiatum*	Lv	L/S/I/FT	10	100	2.5 (1–6)	2.5	
Cestoda		BC/L/S/FT	65	100	16.3	16.3	
 Diphyllobothriidea indet	Lv	BC/L/S/FT	65	100	16.3 (7–34)	16.3	
Hirudinea		MC/G	4	50.0	2.0	4.0	
 Nototheniobdella sawyeri	A	MC/G	4	50.0	2.0 (1–3)	4.0	
Notes:

A, adult; Lv, larval; BC, body cavity; L, liver; S, stomach; I, intestine; P, pylorus; FT, fat tissue; SU, surface; MC, mouth cavity; G, gills; n, total number of parasites; Location, location of the parasite inside the host; P[%], prevalence; MI, mean intensity; I, intensity (in parentheses); MA, mean abundance; H′, diversity index; J′, eveness.

Parasites isolated from stomach and intestine were located in the lumen of these organs. New host records are marked with an asterisk (*).

Parasites belonged to the six taxonomic groups Digenea (four species), Nematoda (four), Cestoda (two), Acanthocephala (one), Hirudinea (three), and Copepoda (one) (Tables 2 and 3; Fig. 2). P. georgianus had the highest number of parasite species with 13 species belonging to six different taxa, followed by C. wilsoni (seven species), while numbers of parasite species of C. gunnari (four), N. ionah (three), and P. macropterus (four) were comparatively low. The predominant species were postlarval diphyllobothriidean cestodes and the larval nematodes Contracaecum osculatum and C. radiatum, which occurred in all five host species with overall prevalences of 35%, 55%, and 45%, respectively.

Figure 2 Relative composition of parasite taxa found in five channichthyid fish species.

Results are based on parasite prevalence. Cestoda and Nematoda were present in all examined fish species. White: Digenea, light grey: Cestoda, medium grey: Acanthocephala, blue-gray: Nematoda, brown: Hirudinea, black: Crustacea.

Stomach content analysis

There were no pronounced differences in diet composition of the five examined fish species (Table 4). A total of 71.25% (n = 57) of fish stomachs contained prey items (C. wilsoni: 69.7%, C. gunnari: 96.0%; N. ionah: 66.7%; P. macropterus: 100%; P. georgianus: 26.7%). Only Crustacea were found among all stomach samples. Due to the frequently advanced state of digestion, only a minor part could be identified to a lower taxonomic level. Prey items that could be further identified belonged to the order Euphausiacea, which constituted an average abundance of 70.2% with a numerical percentage of prey (n) ranging from 37.8 to 90.5%, mostly of the genus Euphausia sp., and an average frequency of occurrence of F[%] = 54.2 ranging from 25.0% to 83.3%. On the species level, Euphausia superba was identified in fish of C. wilsoni and C. gunnari with F[%] = 30.4% and F[%] = 54.2, respectively. Single specimens of the order Amphipoda were only found in C. gunnari, with F[%] = 8.3.

Table 4 Food items isolated from the five channichthyid species.

Fish species	Food item	F[%]	N[%]	W[%]	IRI	
C. wilsoni (n = 33)	Crustacea indet.	60.9	35.3	33.3	4177.7	
Euphausiacea	47.8	57.6	59.0	5573.5	
Euphausia sp.	30.4	39.5	32.9	2201.0	
Euphausia superba	17.4	26.5	29.3	970.9	
C. gunnari (n = 25)	Crustacea indet.	33.3	10.0	10.0	666.0	
Amphipoda	8.3	1.5	0.3	14.9	
Euphausiacea	83.3	90.5	88.6	14919.0	
Euphausia sp.	54.2	62.1	52.8	6227.6	
Euphausia superba	29.2	23.4	35.9	1731.6	
N. ionah (n = 3)	Crustacea indet.	50.0	23.2	8.3	1575.0	
Euphausiacea	50.0	76.8	91.7	8425.0	
Euphausia sp.	50.0	76.8	91.7	8425.0	
P. macropterus (n = 4)	Crustacea indet.	75.0	62.2	58.2	9030.0	
Euphausiacea	25.0	37.8	41.8	1990.0	
Euphausia sp.	25.0	37.8	41.8	1990.0	
P. georgianus (n = 15)	Crustacea indet.	50.0	11.5	8.1	980.0	
Euphausiacea	50.0	88.5	84.0	8625.0	
Euphausia sp.	50.0	88.5	84.0	8625.0	
Note:

F[%], frequency of occurrence; N[%], numerical percentage of prey; W[%], weight percentage of prey; IRI, index of relative importance.

Molecular nematode identification

The 117 obtained sequences using BLASTn-analyses showed highest congruence with eight different sequences that could not all be annotated to a species level in GenBank (Table S3). The vast majority (100 sequences) belonged to the genus Contracaecum with Contracaecum cf. osculatum D (31) and C. cf. osculatum E (28) being the dominant genotypes. C. radiatum (14) and C. osculatum (s.l.) (8) were identified, and 16 individuals that were assigned to Contracaecum sp. Three sequences were assigned to Contracaecum aff. multipapillatum A and one sequence was identified as Pseudoterranova decipiens. Interestingly, 16 sequences could not be assigned to any of the anisakid genera present in the Antarctic and showed highest similarity (approx. 85%) to the ascarid Parascaris equorum.

Phylogenetic analyses

A total of 138 Operational Taxonomic Units were included in the final dataset. In order to further identify the taxonomic status of the anisakid nematode sequences obtained, a set of 18 reference anisakid sequences of known identity (Pseudoterranova krabbei HM147279; P. decipiens (s.s.) HM147278; P. decipiens E HM147282; P. cystophorae EU477209; P. ceticola DQ116435; P. cattani KC782949; P. bulbosa HM147280; P. azarasi HM147281; C. radiatum EU477210, C. osculatum (s.s.) EU477206; C. osculatum E EU477207; C. osculatum D EU477205; C. osculatum B EU477404; C. osculatum A EU477203; C. osculatum baicalensis EU477208; C. ogmorhini (s.s.) EU477211; C. mirounga EU477213; C. margolisi EU477212) were included in the phylogenetic reconstruction (Fig. 3). A. simplex s.s. (KX158869) was used as an outgroup. Most of our sequences clustered with the respective species clade according to the species identification via BLASTn (Fig. 3). The single sequence of P. decipiens (Pg_11_4b) forms a sister clade together with P. decipiens E. A total of six individuals that were identified as Contracaecum sp. were grouped within the same clade as C. mirounga. The three sequences that showed low identity with Contracaecum aff. multipapillatum A clustered together with the 16 sequences of “P. equorum” as a monophyletic sister clade of C. margolisi and C. ogmorhini (s.s.).

Figure 3 Phylogenetic tree of nematode sequences.

Phylogenetic reconstruction of 117 obtained sequences and 19 anisakid reference species. Branch support values are given for the consensus tree constructed from 9,902 original raw trees. For sample abbreviations please see Table S3. The following sample-coding was used: “Species initials_Host-Nr._Nematode-No.”. GenBank accession numbers were given for reference sequences.

Discussion

Overall, 15 metazoan parasite species of 13 different genera were identified. Among the detected parasite species, nine and six species were present as larval or adult stages, respectively, with larval parasites showing high infestation rates. Generally, adult stages would indicate the fish as the final host for the parasite species, whereas larval stages are found in intermediate hosts feeding on zooplankton. P. georgianus showed the highest diversity with 13 different parasite species. This high number of taxa might reflect the feeding behavior, life style and zone preference of the fish, ranging from benthic to pelagic (Iwami & Kock, 1990). The other four fish species had a less diverse fauna. Compared to other members of the suborder Notothenioidei (e.g., Dissostichus eleginoides (47 parasites species), Macrourus whitsoni (25), Notothenia coriiceps (37) (Palm et al., 1998; Klimpel et al., 2009; Münster et al., 2016)) this is a rather low parasite diversity, found within this study (see also Table S1). Among the sampled species, Cestoda (postlarvae) were the dominant parasite groups in terms of prevalence and intensity. Cestoda often use Antarctic elasmobranchs and birds as final hosts (Rocka, 2017). For nematodes, except for Ascarophis nototheniae, Channichthyidae serve as intermediate hosts and marine mammals as final hosts and occurred as L3-larvae in the examined fishes. In the case of the genera Pseudoterranova and Contracaecum, the main final hosts are Pinnipedia (Klöser et al., 1992; McClelland, 2002), just like the only acanthocephalan Corynosoma bullosum, which was isolated from P. georgianus, matures in a pinniped host (e.g., Mirounga leonine) (Zdzitowiecki, 1991a). These findings emphasize the role of Channichthyidae as important intermediate hosts, mostly preying on crustaceans infected with larval stages. The low infection parameters (prevalence and intensity) and diversity of Digenea is surprising as they are probably the most diverse metazoan parasite group within Antarctic waters (Rocka, 2006). Whether seasonal variations of abundances, as assumed to occur in some digenean species (Rohde, 2005), might have influenced these numbers cannot be evaluated due to a lack of comparable studies taken during the austral winter. Another explanation could be the high abundance of Euphausiacea in these waters, which are not known to harbor larval digeneans. The low diversity is in accordance to the findings in previous studies (Table S1).

In the following we are going to discuss the fish according to their ecology.

Champsocephalus gunnari and Chaenodraco wilsoni

A high similarity of C. wilsoni and C. gunnari (81.6%) was revealed and might be explained by the fact that these two species share a similar food spectrum, including almost exclusively Crustacea. C. wilsoni complements its diet with other small crustaceans such as the Euphausiacea Thysanoessa macrura, Amphipoda (Themisto gaudichaudii), and occasionally small fish (Pleuragramma antarctica) whereas C. wilsoni includes also Hyperiidea and Mysida (Iwami & Kock, 1990; Eastman, 1993; Kock et al., 1994; Frolkina, 2002; Flores et al., 2004; Kock, 2005; Kock et al., 2008; Kock, Gröger & Jones, 2013). Stomach content analyses of the present study support previous investigations, with food items mainly consisting of Euphausiacea, which in some cases could be identified as E. superba. No residues of fish species were found, which supports the role of C. wilsoni and C. gunnari as highly specialized on krill. Parasite diversity and infection parameters are also in accordance with the postulated feeding ecology of the species. Parasite species diversity was low and consisted of mainly larval cestodes (Diphyllobothriidea indet.), nematodes (C. osculatum, C. radiatum, Contracaecum sp., Nematoda indet.), and one digenean species (C. gunnari: Macvicaria georgiana; C. wilsoni: Neolebouria antarctica). These parasites use fish as second intermediate or paratenic hosts. M. georgiana and N. antarctica were present in the adult stage, respectively, indicating the role of the fish as final hosts in the life cycle of these digenean parasites.

Digenea are mainly present in coastal zones of Western Antarctica while they are scarce in the open sea and deep waters. Especially in the Weddell Sea and around the South Shetland Islands this parasite taxon is rare and particularly the species N. antarctica shows a low abundance in this area compared to other Antarctic regions (South Georgia) (Zdzitowiecki, 1991b, 1997; Rocka, 2006; Oguz et al., 2012). Representatives of the order Diphyllobothriidea are commonly known to infect zooplankton (i.e., E. superba) as first intermediate host and are frequently found around the South Shetland Islands in channichthyid representatives (Kock, 1992; Rocka, 2003, 2006; Palm, Klimpel & Walter, 2007; Oguz et al., 2012). Due to the fact that exclusively plerocercoid larval stages were present C. gunnari and C. wilsoni can be considered intermediate hosts.

The most prominent parasites in prevalence and intensity were the Nematoda with the larval stages (L3) of the species C. osculatum and C. radiatum which both infect seals (e.g., Leptonychotes weddellii, in this region) as a final host (Klöser et al., 1992). Channichthyids are known to occupy a key position in the life cycle of Antarctic Contracaecum (Kock, 1992; Oguz et al., 2012). The present study reveals a slightly higher prevalence and intensity of C. osculatum than of C. radiatum. This observation is contrary to the assumption, that C. radiatum developed a pelagic life cycle and therefore shows a higher abundance in pelagic fish species. On the contrary, C. osculatum developed a benthic life cycle (Klöser et al., 1992; Kock, 1992; Palm et al., 1994). Contradicting observations of the present study can be explained by the targeted host species. Even if representatives of the Channichthyidae play a key role in the life cycle of Antarctic Contracaecum, both fish might be representatives of lower importance. C. radiatum is highly abundant in piscivorous channichthyids like Chaenocephalus aceratus, C. rhinoceratus, and C. dewitti that feed on pelagic Pleurogramma antarcticum (Klöser et al., 1992; Kock, 1992; Palm et al., 1994).

Neopagetopsis ionah and Pagetopsis macropterus

Reliable data on the feeding behavior of N. ionah and P. macropterus are scarce. The food spectrum is described as a mixture of euphausiids, copepods, pteropods, and fish where the proportion of different items in the diet varies greatly with time and space (Kock, 1992; Eastman, 1993; Kock, 2005; Kock, Gröger & Jones, 2013). Stomach content analyses of the present study revealed only food items belonging to the Crustacea (Table 4). More precisely identified residues were assigned to the order Euphausiacea or the genus Euphausia. No residues of copepods, pteropods or fish were found. The absence of these food items could be explained by the fact that individuals around the South Shetland Islands prey preferential on euphausiids as this taxon is strongly represented and provides a high biomass in this region (Ichii, Naganobu & Ogishima, 1996; Reiss et al., 2008).

Parasitological findings are in accordance with stomach content analyses but contradict literature data which describe N. ionah as piscivorous (Iwami & Kock, 1990; Kock, 1992, 2005; Eastman, 1993; Kock, Gröger & Jones, 2013). Present stomach content and parasitological investigations indicate a diet mainly based on krill and other crustaceans for N. ionah. Compared to the other four channichthyid species, P. macropterus showed higher intensities of the three endoparasites Diphyllobothriidea indet., C. osculatum and C. radiatum. In relation to the literature about the feeding ecology, these higher parasite intensities could be explained by a piscivorous feeding of P. macropterus which leads to an accumulation of parasites via the prey (Kock, 1992; Williams, MacKenzie & McCarthy, 1992; Klimpel, Seehagen & Palm, 2003; Palm, Klimpel & Walter, 2007). Because parasite intensity and diversity integrates the feeding ecology for a longer period, in the case of P. macropterus, it is more likely, that the fish species principally has a diet including fish. However, this could not be verified by stomach content analyses. Thus, it was not possible to contradict the hypothesis of a principally piscivorous diet in the species. The small sample size for both species is not representative in order to interpret the feeding behavior.

Pseudochaenichthys georgianus

Like N. ionah and P. macropterus, the food spectrum of P. georgianus includes Crustacea and fish, with the proportion of the different items varying in time and space (Kock, 1992, 2005; Eastman, 1993; Kock, Gröger & Jones, 2013). Similar to the former two species it is possible, that individuals of P. georgianus around the South Shetland Islands prey intensively on krill because of its high abundance. The problem of contradictions in parasitological findings and stomach content analyses in P. georgianus are similar to P. macropterus.

Endoparasite diversity and intensity of P. georgianus was high and greatly exceeded parasite parameters of the other four fish species. Similar to the previous four species the cestode Diphyllobothriidea indet. and the nematode species C. osculatum, C. radiatum, and Contracaecum sp. were isolated in the larval stage. In addition, high numbers of the cestode Tetraphyllidae indet. and the nematode P. decipies were detected. For both taxa P. georgianus serves as a second intermediate or paratenic host. Final host for Tetraphyllidae indet. are elasmobranchs, while representatives of Pseudoterranova are parasites of marine mammals. High prevalence and parasite intensity in P. georgianus (Table 2) points to a strong accumulation of parasites through a piscivorous or unspecialized feeding behavior. In contrast to the other four channichthyid species, P. georgianus was infected with a species belonging to the acanthocephalans. C. bullosum infects marine fish as intermediate host, mammals or birds act as final hosts (Zdzitowiecki, 1991a). C. bullosum occurred in the cystacanth stage of development in P. georgianus. The life cycle of Acanthocephalans is largely linked to the benthic environment, and the group is rarely found in planktonic organisms (Marcogliese, 1995). Thereby benthic amphipods and ostracods are used as first intermediate hosts. Fish that exploit the bottom layer are therefore commonly used as second intermediate or final host, depending on the acanthocephalan species (Zdzitowiecki, 1991a; Kock, 1992; Rocka, 2006; Palm, Klimpel & Walter, 2007). The occurrence of this parasite in P. georgianus gives therefore more information about the horizontal distribution of its host than about the diet.

Molecular analyses of nematodes

Molecular analyses of a subsample of anisakid nematodes and the subsequent phylogenetic reconstructions were conducted in order to confirm previous morphological identification and identify the nematodes to (sibling) species level.

While GenBank analyses did not provide sufficient resolution for identification to species level (e.g., Contracaecum sp., Nematoda indet.), phylogenetic reconstruction including a set of 18 reference species allowed a more accurate species assignment. Analyses clearly revealed the species C. osculatum (sensu lato), which splits into the two sibling species C. osculatum D and C. osculatum E. Another nematode species, C. radiatum, was identified by molecular investigations (Table S2). No sibling species of C. radiatum are known. The parasite species is known to be distributed in the Southern Ocean and occurs in high numbers around Antarctica. Representatives of the Channichthyidae are also known to be infested with larval forms of C. radiatum (Klöser et al., 1992; Kock, 1992; Arduino et al., 1995; Mattiucci & Nascetti, 2007; Palm, Klimpel & Walter, 2007).

The species P. decipiens (s.l.) was identified which supports previous investigations in the Antarctic region (Palm et al., 1994; Zhu et al., 2002; Palm, Klimpel & Walter, 2007; Timi et al., 2014). Only one sequence was obtained and it was assigned to the species P. decipiens E, a sibling species of the P. decipiens complex known from Antarctic waters. It differs genetically from the four-other species of the complex (P. bulbosa, P. azarasi, P. krabbei, P. cattani) from the North Atlantic, Norwegian Sea, Baltic Sea, and Arctic-Boreal and Japanese waters (Zhu et al., 2002; Mattiucci & Nascetti, 2007; Timi et al., 2014). Previous studies included that it can be assumed that all individuals of P. decipiens found in this study belong to the sibling species P. decipiens E. However, further investigations based on the cox2 marker are needed to verify this assumption. Pseudoterranova was only found in representatives of the host species P. georgianus and could not be detected in any of the other targeted fish species.

A total of 16 nematode samples revealed similar sequences using sequence analysis, but could not be clearly assigned to existing sequences in GenBank (Fig. 3; Table S3). These samples came exclusively from nematodes found in the host species C. wilsoni and C. gunnari. Sequence analyses (BLASTn) showed the highest sequence identity to a phylogenetically related species annotated as P. equorum in GenBank. This equine ascarid nematode is an important cosmopolitan in foals (Lindgren et al., 2008). Species like this or closely related ones have no affiliation to the marine or Antarctic environment in any life stage and were never found in this region. This excludes the occurrence of individuals of this species in representatives of the Channichthyidae. This is also supported by the low sequence identity of the GenBank comparison (83–85%, Table S3). The relevant sequences form a separate clade including also those sequences that have been identified as Contracaecum aff. multipapillatum A before. These results indicate that the sequences might belong to a species of Contracaecum that have not been annotated before. Further investigations are needed in order to clarify their taxonomic status.

Conclusion

Although Channichthyidae have been extensively studied, only few have dealt with the ecology or the parasite fauna. The large number of 15 new host records found in this study shows the importance of studies within Antarctic water to gain a better understanding of this unique habitat. Most recorded parasite species can be characterized by a broad host range including a variety of different Notothenioidei, emphasizing their important role as hosts in the Southern Ocean. Furthermore, a possible undescribed genotype or even species might exist among the nematodes.

Supplemental Information

Supplemental Information 1 Parasite fauna of channichthyid species, based on literature data and own studies.

Species occurring outside of the Antarctic Convergence (e.g. South Georgia Island) are included. Records are based on Klimpel et al. (2009) and Oguz et al. (2015). Depth ranges of the fish species are taken from Froese & Pauly (2016). Abbreviations: D, Digenea; C, Cestoda; N, Nematoda; A, Acanthocephala; Cr, Crustacea; H, Hirudinea.

Click here for additional data file.

Supplemental Information 2 Morphometric data on the host species investigated.

n = sample size, TL = total length, TW = total weight, SW = slaughter weight. Arithmetric mean and standard deviation are given. Median, minimum and maximum are given below. K = condition factor, HSI = hepatosomatic index.

Click here for additional data file.

Supplemental Information 3 Molecular results.

Nematodes subjected to molecular analyses and identity matching with the NCBI data set (GenBank). ID = Sample code: Species initials_Host-No._Nematode-No., Organism = Species of highest identity in GenBank, % Identity: Pairwise identity of query and organism.

Click here for additional data file.

Supplemental Information 4 MUSCLE sequence alignment of the obtained cox2 sequences including species references.

Complete alignment that has been used for subsequent phylogenetic analyzes. Sequence IDs according to Table S3 and Fig. 3. Reference sequences were obtained from Genbank (Accession numbers indicated).

Click here for additional data file.

Supplemental Information 5 Description of raw data including sampling information, morphological measurements and stomach contents.

ID = identification code of the examined fish specimens (C.w: Chaenodraco wilsoni; C.g: Champsocephalus gunnari; N.i: Neopagetopsis ionah; P.m: Pagetopsis macropterus; P.g: Pseudochaenichthys georgianus). Hol = sampling point, SL = standard length, TL = total length, TW = total weight, CW = carcass weight, GO = gonad weight, LW = liver weight, SW = stomach weight, FW = food item weight.

Click here for additional data file.

Supplemental Information 6 Raw data of parasitological examination.

Given numbers are the amount of recorded parasite specimens. ID = identification code of the examined specimens. Hir: Hirudinea, N. saw: Nototheniobdella sawyeri, N. noto: Notobdella nototheniae, T. cap: Truliobdella capitis, Ne: Nematoda, C. osc: Contracaecum osculatum (s.l.), C. rad: C. radiatum, C. sp.: Contracaecum sp., P. dec: Pseudoterranova decipiens (s.l.), A. not: Ascarophis nototheniae, Dig: Digenea, N. ant: Neolebouria antarctica, M. geo: Macvicaria georgiana, G. phy: Gonocerca phycidis, E. oat: Elytrophalloides oatesi, Ces: Cestoda, Db: Diphyllobothriidea, Te: Tetraphyllidae, Ac: Acanthocephala, C. bul: Corynosoma bullosum, Cr: Crustacea, E. ant: Eubrachiella antarctica.

Click here for additional data file.

We thank Karl-Hermann Kock, Volker Siegel, and Christopher Jones for their contributions during the field work on the RV Polarstern and we gratefully acknowledge Birgit Nagel for her technical assistance.

Additional Information and Declarations

Competing Interests

Author Contributions

Data Availability

The authors declare that they have no competing interests.

Thomas Kuhn conceived and designed the experiments, analyzed the data, contributed reagents/materials/analysis tools, authored or reviewed drafts of the paper, approved the final draft.

Vera M.A. Zizka performed the experiments, analyzed the data, prepared figures and/or tables, authored or reviewed drafts of the paper.

Julian Münster conceived and designed the experiments, analyzed the data, prepared figures and/or tables, authored or reviewed drafts of the paper, approved the final draft.

Regina Klapper analyzed the data, prepared figures and/or tables, authored or reviewed drafts of the paper, approved the final draft.

Simonetta Mattiucci analyzed the data, authored or reviewed drafts of the paper.

Judith Kochmann authored or reviewed drafts of the paper, approved the final draft.

Sven Klimpel conceived and designed the experiments, contributed reagents/materials/analysis tools, authored or reviewed drafts of the paper, approved the final draft.

The following information was supplied regarding data availability:

The raw data are provided in the Supplemental Files.

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
