# Peer review of "Lighten up the dark: metazoan parasites as indicators for the ecology of Antarctic crocodile icefish (Channichthyidae) from the north-west Antarctic Peninsula"

_PeerJ, doi:10.7717/peerj.4638_

## Round 0.1 · original submission · Major Revisions

The reviewers indicate that your paper has merit, but they disagree on the final decision - I have opted for major revisions in this case, as Reviewer 2 has clearly spent a considerable amount of time to provide positive feedback, but has some significant concerns I would consider this decision to fall on the 'minor' side of the major revisions category. However, I would ask that you respond fully to reviewers comments before returning the manuscript.

·

Basic reporting

The English usage is generally good, but the punctuation, particularly the use of commas, requires some editorial attention in some sections. The authors show a good knowledge of the literature apart from that on the recent changes in classification of cestodes. The structure of the manuscript conforms to the usual standard sections.

Experimental design

No comment.

Validity of the findings

No comment.

Additional comments

This is overall a well written and presented manuscript and I have only a few comments and suggestions for improvement. They are as follows.

Among the parasite taxa reported are “Pseudophyllidae indet.” The cestode order Pseudophyllidea is no longer recognized; it was suppressed by Kuchta et al. (2008), who proposed two new orders – Bothriocephalidea and Diphyllobothriidea. Considering the sites of infection of these cestodes in the fish, I assume that they were postlarvae, in which case they would have been diphyllobothriideans, maturing in marine mammals. The references in the text (lines 192, 276, 285) and in Table 3 to pseudophyllidean or Pseudophyllidae should therefore be changed to diphyllobothriidean or Diphyllobothriidea respectively. It would also be helpful in Table 3 to refer to both the diyphyllobothriidean and tetraphyllidean cestodes as postlarvae, and to distinguish the nematodes as either larvae or adults. Also in Table 3 Acanthocephala is wrongly spelt. Here is the full reference to the Kuchta et al. paper.
Kuchta, R., Scholz, T., Brabec, J and Bray, R.A. (2008). Suppression of the tapeworm order Pseudophyllidea (Platyhelminthes: Eucestoda) and the proposal of two new orders, Bothriocephalidea and Diphyllobothriidea. International Journal for Parasitology 38, 49-55.

In the Conclusion (line 401), 15 new host records are mentioned for the first time. This information should be given earlier at the beginning of the Results section. The new records should also be indicated in Table 3, for example with asterixes and a footnote.

Lines 239 to 243. This paragraph repeats what has already been stated in the Introduction and is unnecessary.

Line 255. Birds can also be final hosts for the genus Contracaeceum.

Line 350. It seems very unlikely that fish could be final hosts for Corynosoma bullosum. References to records from mammals and birds would be useful.

Congratulations to the authors on a valuable contribution to our knowledge of the diet and parasite faunas of a group of fish about which relatively little is known.

Reviewer 2 ·

Basic reporting

A) The abstract is written in a manner that promotes interest in reading the manuscript. However, the manuscript would benefit from further synthesis and improved clarity in both the writing and some data presentation. The writing is generally good but it is not unambiguous throughout the manuscript. Some sections would benefit from improvements to writing and greater attention to important details. Examples are provide below:
1. Lines 388-389: “However, sequences were assigned to the species Parascaris equorum by comparisons with GenBank.” This statement is not accurate and subsequent writing acknowledges this fact. GenBank does not assign species identification through BLAST. The GenBank BLAST simply provides top BLAST hits based upon sequences available in that database. Assigning these nematodes an identification of Parascaris equorum would be erroneous based on that BLAST analysis. This parasite is clearly not a horse nematode but it is related to that parasite phylogenetically. Subsequent writing acknowledges this fact but some statements are incongruent with this fact if viewed in isolation (i.e., Table S2). The column heading in Table S2 should be changed from Organism (= species identified by GenBank) to Organism of Highest Identity for accuracy and clarity.
2. Lines 194-195: “Tetraphyllidean cestodes occurred at high intensities, with a mean intensity of mI=77.5±63.5 averaging all fish species.” This is misleading as written. According to Table 3, only one of the five host fishes actually harboured Tetraphyllidean cestodes. Sentence structure contributes in part to this particular issue.
3. Line 198: “There were no differences in diet composition of the five examined fish species”. There might have been no statistical differences, but the paragraph that follows this statement highlights several differences that were observed in diet composition.
4. Lines 91-93: This sentence would likely be more effective if re-written to convey more effectively both the current state of knowledge and the significance to the current investigations. Neither of the two papers cited discuss sampling effort and so perhaps you could reposition the citations within that sentence and then add another citation that makes the case for sampling effort. Is an analysis from 80 fish hosts distributed among 5 genera sufficient – justified from other studies?
5. Line 386-388: “Samples came exclusively from nematodes found in the host species Chaenodraco wilsoni and Champsocephalus gunnari.” Isn't this referring to the “Nematode indet” that was also found in the host P. georgianus? Perhaps the single specimen from P. georgianus was not sequenced but it should be indicated that another fish host also had an unidentified nematode – is this likely to be the same parasite and with a broader host range?
6. Lines 146-149: This section provides no indication of what 19 reference species were included, where their sequence data was acquired, or how/why these Anisakid nematodes were selected. Additional details would improve understanding here.
7. Line 157: What is the GenBank Accession number for the sequence that was included as an outgroup for the tree? This isn’t in the text or in the tree. How/why was this species chosen?
8. Lines 427: Insufficient citation. The software producer suggests citing the user manual as follows: Clarke, KR, Gorley, RN, 2006. PRIMER v6: User Manual/Tutorial. PRIMER-E, Plymouth, 192pp.
9. Lines 97-98: It is not obvious how this current study was performed to increase knowledge of parasite life cycles per se but rather it clearly expands our knowledge of potential intermediate/paratenic hosts that could participate in the life cycle. Line 280 makes brief mention of life cycle knowledge acquired in the current study.
10. Lines 39-40: “The stomach content, with only Crustacean (Euphausiacea, Amphipoda) in all examined species, was less diverse.” The sentence structure could be improved – perhaps: “The stomach contents were less diverse with only Crustacea (Euphausiacea, Amphipoda) recovered from all examined fishes.”
11. Line 211-213: “Blastn-analyses partially yielded equivocal results. The 117 obtained sequences showed highest congruence with eight different sequences that were in parts not annotated to a species level in GenBank (Table S2).” Both of these sentences could be improved easily to convey the information more effectively.
12. Line 275: “Parasite species diversity was low and considered mainly…” Writing could be improved here.
13. Line 325: “due to the intake of already infected other fish” Writing could be improved here.
14. Lines 370-371: “Also representatives of the Channichthyidae were known to be infested with larval forms of…” The sentence structure could be improved.
15. Lines 376-378: “Only one sequence was obtained and could be compared with a GenBank reference. It was assigned to the species P. decipiens E which is a sibling species of the P. decipiens complex, known from Antarctic waters.” Merging sentences could improve the writing here, perhaps: “Only one sequence was obtained and it was assigned to the species P. decipiens E, a sibling species of the P. decipiens complex known from Antarctic waters.”
16. Line 378: “It genetically differs…” perhaps change to “It differs genetically…”
17. General: Most Figure, Table, and supplementary Table titles would benefit from more clarity in the writing. Line 27: “the Antarctica” change to “Antarctica”
18. Line 38: “six taxonomic groups Digenea” change to “six taxonomic groups including Digenea”
19. Line 67: “cold adapted” change to “cold-adapted”
20. Line 68 and throughout: “Cheng, di Prisco & Verde, 2009” to “Cheng et al., 2009”
21. Lines 79-80: Eight families stated but only seven listed. Artedidraconidae is missing.
22. Line 80 and throughout: “Near, Pesavento & Chenge, 2003” to “Near et al., 2003”
23. Line 83 and throughout: “Near, Pesavento & Chenge, 2003” to “Near et al., 2003”
24. Line 92 and throughout: “Klimpel, Seehagen & Palm,, 2003” to “Klimpel et al., 2003”
25. Line 94: “content” to “contents”
26. Line 138: “were controlled” perhaps a better descriptor “visualized”?
27. Line 146: “the BLASTn” change to “a BLASTn search”
28. Lines 160-161: Parasite abundances? Parasite intensities?
29. Line 191: “sampling” change to “sample”.
30. Line 194: “which occurred in all five species with overall prevalences” perhaps change to “which occurred in all five host genera/species with overall prevalences of”
31. Line 211: “Blastn” change to “BLASTn”
32. Line 211-218: “were” change to “was”
33. Line 222: Should spell out on first use of the acronym - Operational Taxonomic Units.
34. Line 222: Should indicate that the sequences being compared (study and reference) are Anisakid nematodes.
35. Line 224: Delete period after Pseudoterranova
36. Line 226: Italicize P. azarasi
37. Line 231: “Blastn” change to “BLASTn”
38. Line 242: “content” to “contents”
39. Line 250: Delete one of the double brackets
40. Line 251: “table” change to “Table”
41. Line 256: “Acanthocephala” change to “acanthocephalan”
42. Line 259: “the low infection parameters” It is unclear what this means specifically.
43. Line 264. Italicize C in C. wilsoni
44. Line 280: “role of the fish as final hosts in the life cycle of these parasites.” Perhaps change to “role of the fish as final hosts in the life cycle of these digenean parasites.”
45. Line 288. “Palm, Klimpel, & Walter, 2007” to “Palm et al., 2007”
46. Line 292: Delete “two”
47. Line 321: Should “three” be “four”?
48. Lines 334-335: Generic names can be abbreviated after first use.
49. Lines 340-342: Generic names can be abbreviated after first use.
50. Line 360: “nematods” change to “nematodes”
51. Line 366: “detailed” perhaps change to “accurate”
52. Line 393: “by the low values of sequence identity” perhaps change to “by the low sequence identity”
53. Line 404: “host” change to “hosts”
54. Line 480: Need to italicize genus and species name.
55. Line 541: Need to italicize genus and species name.
56. Multiple citations require the first letter in the journal name components to be in upper case: Line 414, 424, 429, 478, 481, 485, 505
57. Multiple citations require additional information to be an acceptable as a citation. Volume number, page numbers, etc: Lines 429, 430, 446, 454, 456. 463, 478, 548, 553
58. Line 589: Need a colon after “grey”
59. Line 592: Capitalize Table
60. Line 600: “results for the” perhaps change to “identity”
61. Line 605: Capitalize Table
62. Figure 3 title could be more specific regarding anisakid nematodes. Capitalize t in Table S2.
63. Table 1. Figure legend title could be improved: “Information of” change to Information on”; “caught individuals” change to “individuals caught”; Need a period after the species initial in the five column headers to the far right of the Table.
64. Table 2. Figure legend title could be improved: “data of the investigated species” change to “data on the host species investigated”; Tg is actually TW elsewhere – change to TW for consistency; “Arimetric” change to “Arithmetic”; C = condition factor but it is usually abbreviated as Cf or K factor
65. Table 3. Figure legend title could be improved: “Parasitological data of the examined channichthyid Pseudochaenichthys georgianus” change to “Parasitological data from the channichthyid Pseudochaenichthys georgianus”; S = surface but the table uses SU; For worms from stomach/intestine, are these from the lumen or are they embedded in the mucosa/serosa of these tissues?
66. Table 4. Figure legend title could be improved. See suggestion for Table 3. S = surface but should use SU; For worms from stomach/intestine, are these from the lumen or are they embedded in the mucosa/serosa?; Column row “Cestoda”. Column to the right states L/F/T but should be L/FT

B) The authors focus on examination of helminth and ectoparasites of 80 fishes representing five genera of channichthyids. They recovered and identified (morphological/molecular) helminth parasites representing four digeneans, four nematodes, two cestodes, and one acanthocephalan as well as ectoparasites including three leeches and one copepod. These fishes have not been examined extensively for parasites previously and this study presents 14 genera of parasites representing 15/16 species with 15 new host records for these parasites. Partial sequencing of the cox2 gene was used to identify a subset of nematodes and served to identify one new genotype/species of larval nematode. The introduction provides context and the literature is both well referenced and relevant.

C) The manuscript structure conforms to PeerJ standards.

D) All figures are relevant.
1. Figure 2 could be improved for clarity by inclusion of a legend rather than a description of colors/patterns in the text of the figure legend.
2. Figure 3 needs to be revisited. This phylogenetic tree needs substantial attention before it could be published. It requires synthesis to aid the reader in understanding the key contributions without investment of substantial effort. The tree is presented using raw data that could be synthesized to simplify data presentation for ease of understanding. In its current form the reader is required to spend substantial time decoding the parasite lineages and specimen identifications. The specimens represented in the tree from the current study could be presented more simply to facilitate cross-referencing with the supplemental data. As an example, each “Ant” and underscore is unnecessary and simply takes up space with no net benefit for the reader. As per Table S5, ID can be simply host species and number (Cw1, Cw2, etc.) and followed by reference to worm species/number. Repositioning the outgroup to the top of the page would eliminate need for such a long vertical branch. The Figure should be a higher order synthesis of the data. The term reversed should be eliminated from the tree. It is understood that sequences must often be reversed to facilitate DNA sequence comparisons.
3. Tables 1, 3, 4, 5 are relevant to the manuscript. However, Table 2 could be incorporated as a supplementary data file. There is no integration of host morphometrics from Table 2 with any of the data generated regarding diet or parasite diversity.
4. Tables 3 and 4 should include a column that indicates the life stage of each parasite (adult or larval) to facilitate ease of comparison. I believe that the Nematode indet represent larval parasites? The Discussion section would benefit from earlier introduction of the concept of larval versus adult helminths regarding diet and observations on parasite diversity in these fishes.

E) Raw data is included in the form of five files. Specific comments on these Tables is provided below:
Table S1.
1. The currently listed “this study” should probably be changed to “this study – new host record”. It would seem appropriate to add “this study” to the other rows where appropriate – others reported it previously but your study also reports it.
2. Some cestodes are referred to as “indet” but others not. This should be a uniform descriptor throughout this Table.
3. Reference Kock et al., 1984. Journal name should not be upper case.
4. Two Laskowski references need the “p” in Systematic parasitology to be capitalized.
5. The Munoz reference needs Ascarophis to be capitalized.
6. The Rhode et al. 1995 reference needs the Journal name to be corrected – add capital letters.
7. Santora et al., 29013 needs Dis Aquat Org spelled out.
8. Utevsky, 2005 needs Vestn Zool spelled out.
9. Zdzitowiecki and Cielecka, 1996 needs journal page numbers added.
10. Zdzitowiecki et al. 1999 needs the journal name spelled out in full.
11. Zdzitowiecki et al. 1998 needs journal page numbers added.
12. Inconsistent use of et al. in the reference column for Table S1.

Table S2.
1. The Table title needs improvement. “results for the matching with the NCBI data set” is not accurate. The title also needs some description of the terms E value and Accession as per the other descriptions.
2. The ID column information does not easily correlate with data in Table S5. This information should be simplified to match new IDs for use in the phylogenetic tree.

Table S4.
The Table title needs improvement.

Table S5.
1. The Table title is in need of improvement.
2. N. noto should not be italicized for consistency with other abbreviations.

Experimental design

A) The manuscript focuses on primary research that fits within the scope of PeerJ.
B) The research question is defined and focuses on a survey of the parasites and diet of Antarctic fishes to help understand aspects of the ecology of the host fishes (i.e., trophic interactions). This research serves to fill a knowledge gap regarding fishes that have evolved in an extreme environment using parasites to help understand trophic interactions. Accordingly, the data are relevant and meaningful.
C) Methods are generally described with sufficient detail to replicate the work. Examples of deficiencies in methodology are listed below.
1. Line 127-128: From Tables 3 and 4, 625 non-Ascarophis nematodes (597 Anisakids and 28 Nematoda indet) were collected in this study. Lines 120-122 indicate that some nematodes were kept for morphological identification and the remaining were preserved for molecular identification. 117 nematode larvae were ultimately assessed for nematode species identification using DNA sequencing of cox2 but details are lacking regarding how 117 were selected from the population of 625 nematode larvae. Furthermore, 572 of the 625 nematodes have species level identification in Tables 3 and 4. Does the 117 represent a certain number of representatives from each individual fish species? What morphological features enabled the remaining larvae to be identified to species in the absence of DNA sequencing?
2. Lines 132-134: These PCR reaction mix volumes add to more than the 25 ul reaction volume that is reported; 12.5+1.5+1.5+14+2.5 = 42 ul. You should supply the final concentration of dNTPs and you must identify the specific Taq formulation that you used from this specific supplier (proofreading, high fidelity, hot-start, etc.).

Validity of the findings

The authors provide an interesting data set that clearly required considerable time, energy, and expense to collect and synthesize. The overall finding and conclusions seem valid. There are some challenges with data interpretation that represent corresponding challenges inherent in the collection of biological specimens. In this study, 82.5% of the sampled fishes were collected from 25% of the sample sites (sites 7, 8). Furthermore, 72.5% of the sampled fishes were from two of the five species examined. In Line 188: The authors report “a high similarity in parasite fauna of C. wilsoni and C. gunnari (81.6 %)”. Both fish are specialists that feed on invertebrates but they were also collected predominately in one specific region (sites 7/8) that is some 200-250 km from the other six sample sites where the remaining 17.5% of fishes were collected. The study serves to provide some important baseline observations on parasites but higher order analyses are complicated by low host sample sizes in some groups and by regional considerations regarding fish host distribution/samopling. According to Munoz and Cortes (2009), only those fish species with sufficient sample size should be included in a cluster analysis “because low sample sizes can give a parasite spectrum unrepresentative for a host species.” They applied accumulation curves to evaluate the host sample size necessary to have at least 90% of the parasite species represented from a particular fish species. It is understood that the success of catches is beyond the control of the investigators. It is also understood that the complete parasite records for these fishes remains outstanding but the current state of knowledge is very nicely summarized in Table S1. It is appreciated that the authors express that caution should be exercised in interpreting their cluster analysis results (Lines 190-192). A sample size of three fish of one species and four fish of another species is likely insufficient to present an accurate picture of parasite diversity within a host species. In lines 330-331 the authors make the statement that “The small sample size for both species is not representative in order to interpret the feeding behavior”. Given that the helminths are acquired by feeding, the parasite diversity would similarly be not representative. Munster et al. (2016) state that “only parasites unambiguously identified to the species level were used for Bray-Curtis calculations”. Is this true for the current study where species level identifications were lacking for several parasites?

Additional comments

A number of surprising/interesting observations are reported. Lines 248-251: “Low parasite diversity in four of the five fishes examined relative to other channichthyid fishes.” Lines 260-261: “…low diversity of digeneans is also surprising given their diversity within Antarctic waters.” In spite of rather static water temperatures, is there any evidence that time/season of collection can influence the diversity of parasites by a change in food source for Antarctic hosts? All fishes in this study were collected Feb. 21-Apr. 1. In line 309-310 the authors reference changes in prey species in time and space. There was an absence of fish prey in the stomachs of even the known pisciverous host species (line 317-318). Stomach contents and short-lived parasites can be underrepresented in studies where the data is by nature a snapshot in time. Is the home range of these host species restricted? Can these observations/considerations be further synthesized by incorporating additional parasite data existing from these same hosts (select regional data from Table S1)? Parasitological data from previous studies is presented in S1 but doesn't seem to have been integrated into the manuscript. For example, it seems readily apparent that parasite diversity correlates with the number of studies (C. gunnari; ~29 parasite species, ~14 studies). Sample sizes are not presented but perhaps a higher order analysis possible by integrating select data with regional relevance from Table S1?

---

## Round 0.2 · accepted · Accept

I am happy that you have responded to the reviewers' extensive comments diligently, and your paper is now suitable for publication in PeerJ, well done!

#